# The Influence of Negative Life Events on Suicidal Ideation in College Students: The Role of Rumination

**DOI:** 10.3390/ijerph17082646

**Published:** 2020-04-12

**Authors:** Shuai Wang, Huilan Jing, Liang Chen, Yan Li

**Affiliations:** 1School of Marxism, Dalian University of Technology, No. 2 Linggong Road, Ganjingzi District, Dalian 116024, China; wangshuai@dlu.edu.cn (S.W.); ws016@mail.dlut.edu.cn (H.J.); 2Students’ Affairs Department, Dalian University, No. 10 Xuefu Street, Dalian Economic and Technological Development Zone, Dalian 116622, China; 3School of Marxism, University of Science and Technology Liaoning, No. 189 Middle Qianshan Road, Lishan District, Anshan 114051, China; 4Center for Psychological Development, Tsinghua University, No. 30 Shuangqing Road, Haidian District, Beijing 100084, China; sunling@dlut.edu.cn

**Keywords:** negative life events, suicidal ideation, rumination, college students

## Abstract

This study aimed to investigate the influence of negative life events on suicidal ideation in college students and the role of rumination. Using a cluster sampling method, 894 college students were asked to fill out the adolescent life event scale, ruminative response scale, and suicidal ideation scale. The study revealed the following: (1) negative life events, rumination, and suicidal ideation were significantly positively correlated with each other; (2) rumination played a full mediating role in the influence of negative life events on suicidal ideation; and (3) rumination also played a moderating role in the influence of negative life events on suicidal ideation. Under a high level of rumination, negative life events had a significant positive effect on suicidal ideation in college students; however, under a low level of rumination, negative life events did not have a significant effect on suicidal ideation. Rumination played mediating and moderating roles in the relationship between negative life events and suicidal ideation among college students.

## 1. Introduction

Suicidal ideation is a condition where an individual thinks about, considers, or plans to attempt suicide [1]. Suicidal ideation represents an early risk factor before suicide [2] and is also an important psychological activity before death by suicide. The severity of ideation is a direct predictor of suicidal behaviors [3]; 60% of people with suicidal ideation will try to commit suicide within one year [4]. Negative life events are a major source of stress that can impair individuals’ psychological health because they may lead to anxiety, depression, and other negative mood states. Negative life events are important inducing factors of suicidal ideation in adolescents [5,6,7]. The schematic appraisal model of suicide (SAMS) posits that negative life events, such as competition failure, loss, and interpersonal conflict contribute to suicidal ideation [8]. The integrated motivation–volitional model holds that negative life events are the trigger events for individuals’ suicidal ideation [9]. Previous case studies of suicide in college students have shown that, negative life events can lead to negative emotions such as sadness, loneliness, and expecting to die early, which is a direct cause of suicidal ideation [9,10,11,12]. Based on the literature review, we made the following assumption: negative life events have a significant effect on suicidal ideation in college students.

Rumination is a method of coping with negative moods and involves self-focused attention. It is characterized by self-reflection as well as a repetitive and passive focus on one’s negative emotions [13]. Rumination is a typical negative information processing bias that maintains the emotional influence of negative life events. Since rumination is related to increases in psychopathology and general distress [14], clarifying the function of rumination is critical for a better understanding of adjustment after experiencing negative life events. The cognitive model of suicidal behavior [8] suggests that rumination plays a crucial role in the influence of negative life events on suicidal ideation.

Some studies have shown that rumination is a negative predictor of suicidal ideation [15]. That is, (1) rumination plays a mediating role in the influence of negative life events on suicidal ideation [16,17,18]; (2) rumination plays a mediating role in loneliness on suicidal ideation [19]; and (3) rumination plays a mediating role in the state of anxiety [20]. However, Conway and colleagues put forward rumination as a personality trait [21]. Studies indicated that cognitive flexibility was lower in individuals with a high level of rumination compared with those with a low level of rumination [22]. In addition, individuals with a high level of rumination were more likely to suffer from the vicious cycle of negative emotions [23], and they were more likely to have suicidal ideation [24]. Thus, rumination plays a moderating role in the above relationships. Based on the literature review, rumination is likely to play different roles simultaneously in the underlying mechanisms of suicidal ideation. Thus, we have made the following assumption: rumination plays mediating and moderating roles in the influence of negative life events on suicidal ideation in college students.

According to the response styles theory (RST) [25], studies have typically measured ruminative coping using the ruminative responses scale (RRS). There are three factors: self-focused, symptom focused, and focused on the possible consequences and causes of the mood. However, there are mixed results for the factor structure observed in revisions for different cultural versions. For example, in the sample of college students, Robert et al. [26] obtained a three factor models of RRS, including symptom-based rumination, introspection/self-isolation, and self-accusation. Bagby and Parker obtained a two-factor RRS model of symptom-focused rumination and self-focused rumination by studying outpatients with depression [27].

Treynor et al. found support for a two-factor model of rumination after constructing an RRS and unconfounding it with depression content, reflective pondering, and brooding [13]. Han and Yang [28] revised the Chinese version of RRS, supporting a three-factor model, symptom rumination, reflective pondering, and brooding. These results suggest that rumination can differ by culture. Most studies investigating the mechanism between negative life events and suicidal ideation are in the context of western individualistic culture. It is not known whether these conclusions can be applied to the eastern collectivism culture. Therefore, our study focuses on the mechanism of suicidal ideation among college students in the context of Chinese culture, and analyzes the mediating and moderating effects of rumination, which could be of positive significance to develop suicide theory.

## 2. Materials and Methods

### 2.1. Participants

This research was approved by the academic ethics committee of Dalian University of Technology (DUT18RW506L19). A cross-sectional design was used in the study. Using a cluster sampling method, 1200 participants were selected from four universities in Liaoning province, China. The participants were asked to fill out online questionnaires by mobile phone and were able to opt out voluntarily at any time during the test. The test lasted about twenty minutes. Validation techniques resulted in omitting a total of 117 invalid questionnaires based on unreasonable item response times (≥2 s), 97 for consistent answer responses, and 92 based on the hiding scale (≥4) of self-rating idea of suicide scale [29]. A total of 894 effective samples were used in the analysis, with a valid rate of 74.5%. As shown in Table 1, among the participants, there were 472 males and 422 females; 549 participants were majoring in science, and 345 subjects were liberal arts students, with an average age of 18.88 and standard deviation (SD) was 1.01.

### 2.2. Research Tools

#### 2.2.1. Adolescent Self-rating Life Events Checklist (ASLEC)

ASLEC was developed by Liu et al. [30]. This scale contains 27 items using a 5-point Likert scale, ranging from 0 (never) to 5 (extremely severe impact). The scale covered six subscales: interpersonal relationships (five items; e.g., to be misunderstood or misjudged), learning stress (five items; e.g., failure or dissatisfaction in an exam), punishment (seven items; e.g., to be criticized or punished), loss (three items; e.g., something stolen or lost), health adaptation (four items; e.g., significant changes in diet or rest habits), and other events (four items; e.g., often laments the difficulty of life). In a previous study, the Cronbach’s α of ASLEC was 0.84, and the Spearman–Brown calibration parity half-reliability coefficient was 0.88 [31]. In the present study, the Cronbach’s α of the scale was 0.85 and the subscales’ Cronbach’s αs ranged from 0.76 to 0.89. The fitting indices of confirmatory factor analysis (CFA) were *χ^2^/d f* = 2.96 (*p* < 0.001), goodness-of-fit index (GFI) = 0.989, normed fit index (NFI) = 0.986, incremental fit index (IFI) = 0.989, comparative fit index (CFI) = 0.989, Tacker-Lewis index (TLI) = 0.980, and root mean square error of approximation (RMSEA) = 0.056, suggesting that ASLEC has good construction validity. 

#### 2.2.2. Ruminative Response Scale (RRS)

RRS was developed by Nolen–Hoeksema [32] and revised by Han et al. [28]. This scale contained 22 items and used a Likert 4-point scoring method, ranging from 1 (never) to 4 (always). The scale uses three subscales including symptom rumination (12 items; e.g., I often think: I’m very lonely), brooding (five items; e.g., I often wonder why I always do this) and reflective pondering (five items; e.g., I often write down what I’m thinking about and analyze it). In a previous study, the Cronbach’s α of RRS was 0.90, and the retest reliability was 0.82 [28]. In the present study, the Cronbach’s α for the scale was 0.83 and the Cronbach’s αs of the subscales ranged from 0.78 to 0.92. CFA showed that the RRS has good construction validity (*χ^2^/df =* 2.740, *p* < 0.001, GFI = 0.968, NFI = 0.972, IFI = 0.981, CFI = 0.981, TLI = 0.968, and RMSEA = 0.043). 

#### 2.2.3. Self-rating Idea of Suicide Scale (SIOSS)

The SIOSS was developed by Xia [33]. This scale contains 26 items and uses a 2-point scoring system, ranging from 0 (no) to 1 (yes). Reverse scoring was adopted for 10 items. The scale covered four subscales: hopelessness (12 items; e.g., I often feel pessimistic and disappointed), pessimism (4 items; e.g., I think my life is a failure), sleep (4 items; e.g., I can’t sleep well and easily woken up), and hiding (5 items; e.g., Sometimes I gossip), and hiding (5 items; e.g., sometimes I gossip). Among them, hiding was only used as an indicator of lie detection (≥4) and not included in the statistical analysis. In a previous study, Cronbach’s α of SIOSS was 0.86, the Spearman–Brown calibration parity half-reliability coefficient was 0.81, and the retest reliability was 0.87 [33]. In the present study, the Cronbach’s α of scale was 0.78, and Cronbach’s αs of the subscales ranged from 0.71 to 0.87. The fitting indices of confirmatory factor analysis (CFA) were *χ^2^/df* = 3.152 (*p* < 0.001), GFI = 0.986, NFI = 0.965, IFI = 0.974, CFI = 0.974, TLI = 0953, and RMSEA = 0.052, indicating that the structural validity of SIOSS was good.

### 2.3. Statistical Processing and Analysis

SPSS 19.0 (IBM Corporation, New York, USA), Amos 21.0 (IBM Corporation, New York, USA), and Jamovi [34] were used for statistical processing. The structural validity of the scale was tested by confirmatory factor analysis. Common method biases were tested by Harman’s univariate analysis. The Pearson product–moment correlation coefficient was used to analyze the correlations between the variables. A bias-corrected nonparametric percentile bootstrap method was used to test for the mediating effect. The unconstrained approach was selected to estimate the moderating effect, and a simple slope test was conducted using Jamovi software. 

## 3. Results

### 3.1. Test for Common Method Biases

Common method biases were tested using Harman’s univariate analysis [35]. A total of 21 factors with eigenvalues above 1 were identified, accounting for 31.43% of the total variation, which was smaller than the criterion of 40% [36]. This indicated an absence of severe common method biases in the present study.

### 3.2. Correlation Analysis between Negative Life Events, Rumination and Suicidal Ideation

The mean scores of variables, descriptive statistics, and Pearson product–moment correlation coefficients are shown in Table 2. Negative life events had a significant positive correlation with rumination (r = 0.47, *p* < 0.01), they had a significant positive correlation with suicidal ideation (r = 0.40, *p* < 0.01); and rumination had a significant positive correlation with suicidal ideation (r = 0.66, *p* < 0.01).

### 3.3. Analysis of the Mediating Role of Rumination in the Influence of Negative Life Events on Suicidal Ideation in College Students

Amos 21.0 (IBM Corporation, New York, USA) was used to establish the structural equation of potential variables to test for the mediating effect of rumination in the influence of negative life events on suicidal ideation in college students. The results are shown in Figure 1. This model had good fitness indicators (*χ^2^/df* = 4.43, *p* < 0.001, GFI = 0.96, NFI = 0.95, IFI = 0.67, CFI = 0.96, TLI = 0.95, and RMSEA = 0.06).

Based on the fitting results of the mediation model, the bias-corrected nonparametric percentile bootstrap method was used to test for the mediating effect. Here, the repeated sampling was performed 2000 times [37]. The test showed that negative life events had a significant prediction coefficient on rumination (β negative life events → rumination = 0.55, *p* < 0.001), and rumination had a significant prediction coefficient on suicidal ideation (β rumination → suicidal ideation = 0.79, *p* < 0.001). However, negative life events had an insignificant prediction coefficient on suicidal ideation (β negative life events → suicidal ideation = 0.06, *p* > 0.05). As judged according to the mediation test procedures proposed by Wen et al. [37], rumination played a full mediating role in the influence of negative events on suicidal ideation in college students. The bootstrap method was further used to test for the mediating effect of rumination. The results showed that the indirect effect of rumination was 0.43, with a confidence interval of (0.37, 0.51), excluding zero. This indicated that the full mediating effect of rumination was reliable.

### 3.4. Analysis of the Moderating Role of Rumination in the Influence of Negative Life Events on Suicidal Ideation in College Students

The moderating effect of rumination was tested using the unconstrained approach for potential variables proposed by Wu et al. [38]. In the first step, all variables were normalized; thereafter, the observation indicators of the moderating term were constructed between rumination and negative life events according to the pairing principle for inconsistency in the number of observation indicators described by March et al. [39]. Thus, three indicators of interaction terms were constructed, and the model of moderating effect was constructed, as shown in Figure 2. The results suggested that the model had good fitness indicators (*χ^2^/df* = 3.96, *p* < 0.001, GFI = 0.95, NFI = 0.93, IFI = 0.95, CFI = 0.95, TLI = 0.94, and RMSEA = 0.05). The test results showed that rumination had a significant effect on suicidal ideation in college students (β rumination → suicidal ideation = 0.76, *p* < 0.001); negative life events did not have a significant predicting effect on suicidal ideation (β negative life events → suicidal ideation = 0.05, *p* > 0.05); and the interaction term between rumination and negative life events had a significant predicting effect on suicidal ideation (β negative life events × rumination → suicidal ideation = 0.10, *p* < 0.01). Based on the results above, it was demonstrated that rumination was a significant moderator of the relationship between negative life events and suicidal ideation in college students.

To further analyze the moderating effect of rumination, the subjects were further divided into two groups: a high rumination group (*Mean + 1SD*) and a low rumination group (*Mean − 1SD*). A simple slope test was performed using Jamovi software to analyze the moderating role of rumination. The test results are shown in Figure 3. We found that under a high level of rumination, negative life events had a significant positive predicting effect on suicidal ideation in college students (*Simple slope* = 0.03, *t* = 5.34, *p* < 0.001), but under a low level of rumination, such a predicting effect was insignificant (*Simple slope* = 0.003, *t* = 0.524, *p* > 0.05).

## 4. Discussion

Our study showed that negative life events positively predicted suicidal ideation in college students, which is consistent with previous research [16,40]. Negative life events are direct inducing factors of negative mood states and the primary stress sources of individuals’ suicidal ideation. College students are still not fully developed mentally and may suffer more from negative emotions such as anxiety and depression in the face of competition failure, loss, love-life setbacks, interpersonal conflict, late payments of monthly bills, and academic pressure [41]. These negative emotions will further lead individuals to have more negative cognitions and evaluations of other life events. The research shows that college students’ ability to resolve negative events is relatively weak [42]. When individuals cannot quickly resolve or avoid predicaments due to limited social resources, they tend to doubt that they can solve negative life events or tolerate the negative emotions arising from the negative life events. As a result, they gradually develop a false idea that suicide may be the only solution for current negative mood states, thereby experiencing suicidal thoughts.

Our study showed that rumination played a full mediating role in the influence of negative life events on suicidal ideation in college students. This result was consistent with those of existing studies, which found that rumination was an important cognitive risk factor of suicidal ideation. In a previous study by Smith et al. [43], the presence and duration of suicidal ideation were predicted prospectively by rumination and hopelessness among undergraduates.

Another study suggested that rumination is a risk factor of suicidal ideation and that it can independently positively predict suicidal ideation in female professionals [44]. To be specific, rumination refers to the condition in which individuals suffer from brooding about the impact of negative life events and the associated negative emotions [45], which will further suppress the motivation to solve the problems. Therefore, rumination is an important factor mediating the influence of negative life events on suicidal ideation in college students. 

Other possible consequences include lowering the cognitive threshold that triggers cognitive narrowing, and lack of ability to withdraw, and hence hopelessness. Finally, some individuals may experience suicidal ideation as an escape from their current abject state. Other studies have also shown that individuals who are ruminating on negative life events can be more prone to suicide [18]. The greater the influence of negative life events and the higher the level of rumination, the higher the probability of suicidal ideation [46]. The Activating events-Belief-Consequence (ABC) theory of emotion posits that an activating event affects mood and behavior via a change in cognitive evaluation of the activating event [47].

Our results support this theory: that is, negative live events effect rumination on suicidal ideation, and rumination plays a full mediation role in the relationship between the two variables. Concurrently, another notable theoretical model to explain suicidal ideation, the cognitive model of suicidal behaviors (CMSB), represents the three main constructs that underlie suicidal behavior from a cognitive perspective [48]. CMSB views cognitive processes associated with psychiatric disturbance and suicide-relevant cognitive processes as being activated in the context of life stress. 

CMSB also argues that additional life stress is usually necessary for cognitive processes associated with psychiatric disturbance to activate suicide-relevant cognitive processes, which culminates in the suicidal act [48]. Rumination could make individuals focus on maladaptive cognitive content and negative information processing bias, which leads to thinking that suicide is the only strategy to solve a problem. Thus, rumination as a maladaptive cognitive response style may increase vulnerability to thinking about and attempting suicide ideation by facilitating the activation of a suicide-related schema [17].

We also found that rumination was a significant moderator of the influence of negative life events on suicidal ideation in college students. Specifically speaking, under a high level of rumination, negative life events were more likely to accelerate the emergence of suicidal ideation in college students, which was consistent with the existing research [49]. Evidence suggests that the main characteristics of rumination are repetitive and intrusive thoughts [50], and rumination can lead to depression and extreme anxiety [51]. Confronted with negative life events, college students may lapse into brooding about their impact and the associated negative emotions. Therefore, college students with a high level of rumination are very likely to experience suicidal ideation. 

On the contrary, those with a low level of rumination can effectively control their thinking logic before the emergence of suicidal ideation and thus better cope with the negative life events. Therefore, these college students are more able to reduce the degree of hopelessness and negative emotion, which helps control suicidal ideation. The results also verify the resource model of self-control [52], which says that an individuals’ self-control ability relies on the acquisition of self-control resources. However, the resources available for individuals’ self-control are limited. Under a high level of rumination, a large number of such resources will be consumed, leading to the failure of self-control and a higher probability of suicidal ideation. In contrast, under a lower level of rumination, a smaller amount of self-control resources will be consumed, which increases the probability of the success of self-control and hence suppresses suicidal ideation.

Combining the mediating and moderating roles of rumination, we believe that rumination is not only a trait factor but also a cognitive factor in individuals. These two aspects of rumination fulfill these functions independently and reinforce each other. To our knowledge, there is no study that investigates both the mediating and moderating roles of rumination in the influence of negative life events on suicidal ideation in college students. Therefore, this result needs to be explained theoretically. The differential activation theory of suicidality (DATS) posits that sadness, negative beliefs, and cognitive biases caused by negative life events are one connection with suicidal ideation [53]. 

Rumination can trigger a variety of negative emotional reactions in individuals, such as depression and sadness [54,55,56]. Most research results consistently show that rumination is a susceptibility factor of negative cognitive bias [57]. Therefore, rumination will mediate the relationship between negative life events and suicidal ideation. At the same time, rumination as a trait factor could moderate the relationship between negative living events and suicidal ideation. The results can also be explained by the schematic model of suicide (SAMS) [8]. SAMS posits that negative information processing bias plays an important role in maintaining negative emotional states and obtaining suicidal information.

By experiencing negative life events that are focused on gathering negative information and generating bad moods, individuals with low-level rumination may interrupt their ruminative focus on negative emotions and symptoms, thereby reducing or eliminating any negative effects of rumination on the implications of negative life events for suicidal ideation. However, people with high-level rumination may have attenuated risk when negative life events happen from initiating suicidal ideation. Rumination is a typical negative information processing bias that could maintain the emotional influence of negative living events. Regardless of what happens, individuals with high-level rumination could continue seeking negative information and maintain negative emotions for long periods of time, so that the risk of suicidal ideation increases. Suicide is considered an escape from an intolerable negative emotional state. Therefore, rumination can influence the method of suicide assessment through this moderating role.

Our findings have practical value in preventing university students from committing suicide. On the one hand, college educators can investigate the negative life history of freshmen through questionnaires and interviews. Students with more negative life events may need more ways to cope with stress and frustration. On the other hand, college educators can regard rumination as a risk factor of college students’ suicide for planned interventions. Through education regarding cognitive strategies, college educators could help students reduce negative cognitive strategies and develop positive cognitive strategies, to effectively alleviate the impact of negative life events on suicidal ideation. In addition, colleges should also provide more psychological counseling and group counseling services for students with greater negative life events and higher rumination levels.

However, our study has certain limitations. This was a cross-sectional study, and therefore we could not achieve dynamic tracking of the participants, despite the potential deviations of the self-reporting method used in this study. In the future, a longitudinal study design can be adopted to further track the dynamic relationship between negative life events, suicidal ideation, and rumination. Moreover, the SIOSS questionnaire used in this study does not directly measure suicidal ideation; rather, it measures predictive factors of suicidal ideation, which also leads to limitations. We will select scales directly measuring suicidal ideation to further test the validation of the model in this study. Whether there are other mediating and moderating mechanisms in the influence of negative life events on suicidal ideation in college students remains to be further investigated.

## 5. Conclusions

Our study showed that negative life events, rumination, and suicidal ideation are correlated with each other. The negative life events positively predicted suicidal ideation in college students, and rumination played mediating and moderating roles between them.

## Figures and Tables

**Figure 1 ijerph-17-02646-f001:**
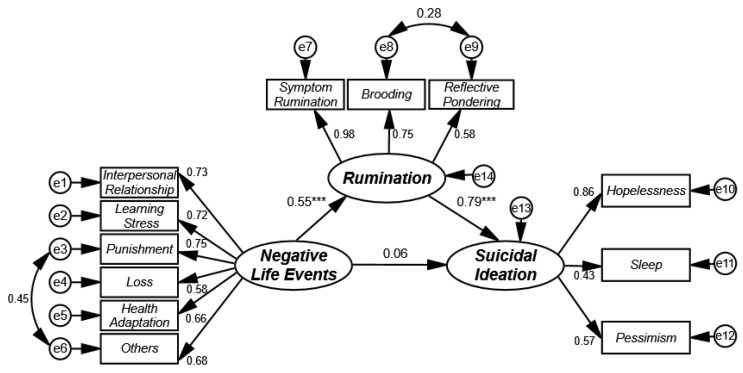
The model of the mediating effect of rumination in the influence of negative life events on suicidal ideation in college students. *** *p* < 0.001.

**Figure 2 ijerph-17-02646-f002:**
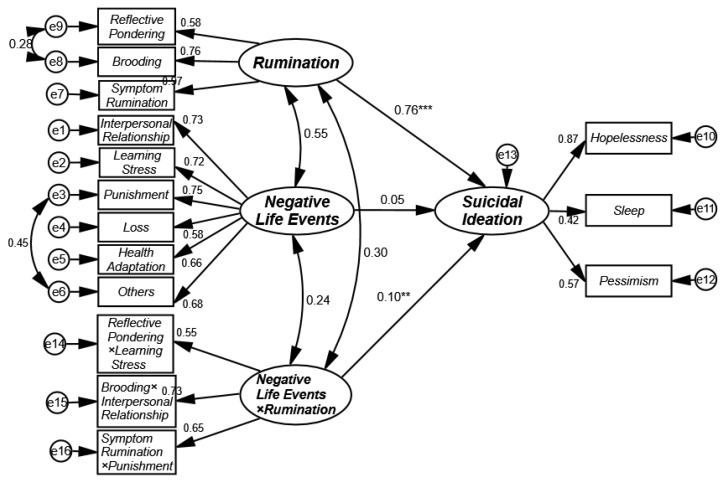
The model of the moderating effect of rumination in the influence of negative life events on suicidal ideation in college students. ** *p* < 0.01, *** *p* < 0.001.

**Figure 3 ijerph-17-02646-f003:**
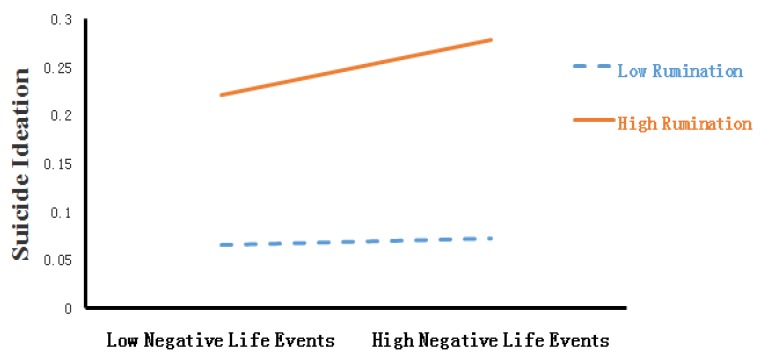
A simple slope test for the analysis of the moderating effect of rumination.

**Table 1 ijerph-17-02646-t001:** Demographics of respondents (*n* = 894).

Variable		Frequency	Percent
Gender	Male	472	52.80%
	Female	422	47.20%
Age	18	387	43.29%
	19	242	27.07%
	20	265	29.64%
Major	Liberal Arts	345	38.59%
	Science	549	61.41%
Source	Rural	368	41.16%
	Urban	526	58.84%

**Table 2 ijerph-17-02646-t002:** Correlation matrix between negative life events, rumination, and suicidal ideation (*n* = 894).

Variable	M ± SD	1	2	3	4	5	6	7	8	9	10	11	12	13	14
**Negative Life Events**															
1. Interpersonal relationship	1.88 ± 0.86	1													
2. Learning stress	1.62 ± 0.83	0.54**	1												
3. Punishment	0.83 ± 0.71	0.55**	0.55**	1											
4. Loss	1.55 ± 1.17	0.41**	0.41**	0.47**	1										
5. Health adaptation	1.03 ± 0.77	0.43**	0.46**	0.51**	0.40**	1									
6. Others	1.23 ± 0.93	0.51**	0.43**	0.72**	0.39**	0.47**	1								
7. Mean scores of negative life events	1.37 ± 0.67	0.77**	0.75**	0.85**	0.66**	0.69**	0.78**	1							
**Rumination**															
8. Symptom rumination	1.79 ± 0.44	0.44**	0.39**	0.32**	0.26**	0.38**	0.37**	0.47**	1						
9. Brooding	2.13 ± 0.53	0.40**	0.37**	0.25**	0.20**	0.31**	0.27**	0.40**	0.73**	1					
10. Reflective pondering	2.00 ± 0.54	0.28**	0.28**	0.22**	0.19**	0.30**	0.20**	0.32**	0.56**	0.58**	1				
11. Mean score of rumination	1.92 ± 0.43	0.44**	0.41**	0.32**	0.26**	0.39**	0.35**	0.47**	0.93**	0.86**	0.77**	1			
**Suicidal Ideation**															
12. Hopelessness	0.19 ± 0.19	0.35**	0.31**	0.25**	0.21**	0.32**	0.28**	0.38**	0.69**	0.55**	0.36**	0.65**	1		
13. Sleep	0.23 ± 0.27	0.26**	0.21**	0.16**	0.18**	0.28**	0.19**	0.27**	0.37**	0.28**	0.23**	0.36**	0.33**	1	
14. Pessimism	0.08 ± 0.16	0.20**	0.11**	0.13**	0.07*	0.23**	0.19**	0.20**	0.44**	0.26**	0.17**	0.37**	0.51**	0.23**	1
15. Mean scores of suicidal ideation	0.17 ± 0.15	0.38**	0.31**	0.25**	0.22**	0.37**	0.30**	0.40**	0.70**	0.53**	0.37**	0.66**	0.92**	0.61**	0.67**

Note: **p* < 0.05, ***p* < 0.01, M = mean, SD = standard deviation.

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
