# Peer review of "The Influence of Negative Life Events on Suicidal Ideation in College Students: The Role of Rumination"

_ijerph, 2020, doi:10.3390/ijerph17082646_

Round 1
Reviewer 1 Report
The study Influence of Negative Life Events on Suicidal Ideation in College Students: The Role of Rumination was performed in Liaoning , China. The main weakness of the study regards the novelty, I'm not sure what is being added to the literature.
There are already studies showing the relation between knowledge about this questions as (to name ones):
Stress-Related Symptoms and Suicidal Ideation: The Roles of Rumination and Depressive Symptoms Vary by Gender
https://www.ncbi.nlm.nih.gov/pmc/articles/PMC5042150/
Subtypes of Rumination in the Relationship Between Negative Life Events and Suicidal Ideation https://www.tandfonline.com/doi/abs/10.1080/13811110902835015?mobileUi=0&journalCode=usui20
Other issues:
Material and Methods: In this section, you need to clearly describe how individuals were approached, how many were approached, how many were eligible, consented or refused. Also, Inclusion and exclusion criteria should be cited with references and cited guidelines for cross sectional observational study may be recommended in order to improve the quality of the manuscript.
The Discussion section is a rehashing of the results. It does not appear that the authors include much interpretation of what the study findings mean for clinical practice or research.
FInally, the conclusión is weak and too long.
I consider that the study is not ready for publication and regret that the disposition is not favorable, but would like to thank you for your support.
We wish you all the best.
Reviewer 2 Report
The authors addressed my comments satisfactorily, either directly, or by including the fact in the Limitations.
However, the sentence not included in the Limitations on line 337 about the number of events per variable is not correct. Specifically, the sentence now reads:
"In terms of analysis, lower number of events per covariate are needed to obtain reliable estimates of regression coefficients when fitting a logistic regression model"
Here, "lower" should be "higher"- more events per covariate are needed than is the case in the present study. Please correct.
Reviewer 3 Report
The authors of this manuscript sought to examine the role of negative life events and rumination on suicidal ideation in a sample of Chinese college students. Given the importance of diversifying the field of psychology and testing existing psychological theories in other cultures, we believe that this study conducted by the authors is valuable and worth publishing. Our enthusiasm for this article, however, is dampened by the language use and grammatical issues found in this article which made the manuscript difficult to follow and understand. We strongly believe, however, that it is important to be inclusive and supportive in our publishing practices as we desperately need to more psychological research that has been conducted in countries where English is not the official language. As such, we hope that the editor will not dismiss this article based on this concern alone. Below are some suggestions and issues that I would like the authors to address before we recommend it for publication:
We struggled with several of the terms used by the authors in this manuscript.
- Given that the field is moving away from using the term “committing suicide,” we hope that the authors will consider using “attempted suicide” or “died by suicide” instead
- The authors described rumination as playing a “regulatory role.” Our understanding is that the authors are suggesting rumination amplifies the relationship between negative life events and suicidal ideation. Given that the authors use the term, “mediating,” in their second aim, we would suggest referring using the term “moderating” in their third aim given that the two are often recognized together.
- The author discussed their results using causal language (e.g. X was a significant predictor of Y). Given that this study uses a cross-sectional design, it would not be possible to test for causal relationships. As such, we hope the authors would adjust their language to be clear that they are not inferring causality (e.g. X had a significant effect on Y).
- In the discussion section, the authors stated that “It is proven again by our study that rumination is indeed an important cognitive risk factor…” Given that we are using this sample to make inferences to the population and many of the variables cannot be directly measured , we cannot prove or conclude anything. As such, we hope that the authors will soften their language and discuss their findings as being supportive or consistent with past studies that have found rumination to be an important risk factor.
Methods:
- We would like the authors to clearly state whether the study uses a cross-sectional or longitudinal design.
- Please provide more information as to what each questionnaire or survey was used to measure and provide, if possible, the psychometric properties, particularly validity, of each questionnaire found in previous studies.
- For the translated and revised version of the RRS, can the authors please explain what symptomatic rumination, obsessive thinking, and retrospection mean in the context of rumination? Also, were these subscales of this RRS and if so, why did the authors choose to combine the three dimensions into one variable instead of keeping them separate?
- For the SIOSS, I’m wondering if there is a question directly measuring suicidal ideation. While hopelessness, optimism, sleep, and hiding may be related to suicidal ideation we cannot say for sure that individuals who are exhibit high levels in each dimensions will have suicidal ideation. If there is no question directly asking about suicidal ideation, then I’m afraid that we cannot say that this instrument measures suicidal ideation. This would be a limitation that needs to be addressed in the discussion.
- We are unsure what the authors mean when they say that “Questionnaire validity was tested by CFA method” and why this was necessary. We would like the authors to provide more clarification and rationale for using a CFA.
Results
- on page 6, line 10 “rumination was a significant predictor of rumination,” I think the authors meant to say that rumination was a significant predictor of suicidal ideation.
Discussion:
- The authors provided the definition of rumination in the discussion section but not in the introduction. Given that their study focuses on the role of rumination, we believe that rumination should be defined and described in more detail in the introduction.
- Given that the authors tested rumination as both a mediator and moderator in the relationship between negative life events and suicidal ideation, I would like to see the authors discuss how the two roles fit together in this context.
- Can the authors provide more clarification on how their results agree with the ABC theory of emotion and S-O-R model?
- It would be helpful for the authors to provide some suggestion on strategies or interventions that can help manage rumination.
Round 2
Reviewer 1 Report
The paper has much improved, and although I have reservations about the interpretation of the data, and the strength of evidence for the clinical message, I think the article presents the data well enough for readers to judge themselves. I would recommend publication.
Reviewer 3 Report
I believe that the authors have addressed my concerns and the manuscript is now ready for publication.